# Joint Effects of Exercise and Ramadan Fasting on Telomere Length: Implications for Cellular Aging

**DOI:** 10.3390/biomedicines12061182

**Published:** 2024-05-27

**Authors:** Shamma Almuraikhy, Maha Sellami, Khaled Naja, Hadaia Saleh Al-Amri, Najeha Anwardeen, Amina Aden, Alexander Dömling, Mohamed A. Elrayess

**Affiliations:** 1Biomedical Research Center, Qatar University, Doha P.O. Box 2713, Qatar; salmuraikhy@qu.edu.qa (S.A.); khaled.naja@qu.edu.qa (K.N.); n.anwardeen@qu.edu.qa (N.A.); 2Groningen Research Institute of Pharmacy, Drug Design, Groningen University, 9712 CP Groningen, The Netherlands; a.s.s.domling@rug.nl; 3Physical Education Department (PE), College of Education, Qatar University, Doha P.O. Box 2713, Qatar; msellami@qu.edu.qa (M.S.); ha1402415@student.qu.edu.qa (H.S.A.-A.); 4Hamad Medical Corporation (HMC), Doha P.O. Box 3050, Qatar; aaden2@hamad.qa; 5College of Medicine, QU Health, Qatar University, Doha P.O. Box 2713, Qatar

**Keywords:** telomere length, human aging, biomarkers, Ramadan fasting, exercise

## Abstract

Aging is a fundamental biological process that progressively impairs the functionality of the bodily systems, leading to an increased risk of diseases. Telomere length is one of the most often used biomarkers of aging. Recent research has focused on developing interventions to mitigate the effects of aging and improve the quality of life. The objective of this study was to investigate the combined effect of exercise and Ramadan fasting on telomere length. Twenty-nine young, non-obese, healthy females were randomized into two groups: the control group underwent a 4-week exercise training program, and the second group underwent a 4-week exercise training program while fasting during Ramadan. Blood samples were collected, and measurements of clinical traits, cytokines, oxidative stress, and telomere length were performed before and after intervention. Telomere length increased significantly from baseline in the exercise-while-fasting group, but showed no significant change in the exercise control group. This increase was accompanied by a reduction in TNF-α, among other cytokines. Additionally, a significant positive correlation was observed between the mean change in telomere length and HDL in the exercise-while-fasting group only. This study is the first to report an increase in telomere length after combining Ramadan fasting with training, suggesting that exercising while fasting may be an effective tool for slowing down the aging rate. Further studies using larger and more diverse cohorts are warranted.

## 1. Introduction

The process of aging is complex, gradually diminishing the functionality of various bodily systems and elevating the susceptibility to developing diseases [1]. Biomarkers of aging are important for understanding the aging process and predicting health outcomes [2]. Telomere length (TL) has been recognized as one of the most often used biomarkers of aging in epidemiological and clinical studies [3]. Telomeres are repetitive sequences located at the ends of chromosomes and play a crucial role in genomic stability and preventing chromosomal degradation [4]. Telomere shortening is a well-known hallmark of both cellular senescence and organismal aging [5]. Aging can be affected by various factors, including lifestyle. Recent research has focused on developing interventions and treatments to mitigate the effects of aging and improve the quality of life. Adopting a healthy diet and maintaining physical activity can significantly influence the aging process and contribute to overall well-being [6].

Diet is an essential factor that affects aging and health. Intermittent fasting is showing promise in improving health and metabolism processes. A large body of research has shown several benefits associated with intermittent fasting, including weight loss, improved heart health, reduced risk of type 2 diabetes, improved blood pressure, and enhanced cognitive functions [7]. Furthermore, intermittent fasting has been found to reduce oxidative damage and inflammation in the body, potentially offering benefits against aging and the development of numerous diseases [8]. Fasting has been practiced for millennia, and many religious groups incorporate periods of fasting into their rituals. Ramadan fasting is one type of intermittent fasting. People engaged in Ramadan fasting must abstain from food, drink, and smoking from dawn to dusk.

Exercise is considered an effective strategy to enhance longevity and health span via its multisystem antiaging effects [9]. Elite athletes usually live considerably longer than the general population [10], and people who engage in physical activity have a significantly lower risk of dying during the same time frame compared to the sedentary group [11]. It is important to note that exercise follows a J-shaped curve, where too much exercise can potentially have negative effects on overall health and lifespan [12]. The effects of exercise on aging occur at both a clinical and cellular level [13]. It can attenuate the major hallmarks of aging, including age-related declines in physical fitness, muscle function, and metabolism [14]. Studies suggested that exercise influences aging through telomere biology by increasing telomerase activity [15].

The relationship between exercise, fasting, and telomere length is a topic of significant interest and ongoing research. While previous studies suggested that exercise and fasting may independently contribute to changes in telomere length, our objective is to investigate the combined effect of exercise and Ramadan fasting on telomere length. To accomplish this, we conducted a longitudinal study aimed at assessing the collective impact of moderate intensity exercise and fasting on telomere length, and we compared that impact to the effects of exercise intervention alone within a cohort of healthy, young, non-obese females.

## 2. Materials and Methods

### 2.1. Study Participants and Selection

Twenty-nine young (20–30 years old) lean or overweight (20 ≤ BMI < 30) healthy females were recruited among Qatar University students. This study was approved by the Qatar University Institutional Review Board (QU-IRB 1798-EA/23). All participants provided informed consent and underwent an initial medical examination to detect any potential underlying health conditions and to ensure their eligibility to participate in the training program. Participants with a BMI less than 20 or greater than 30 kg/m^2^, those with chronic diseases or injuries, and participants who were taking medications prior to the study were excluded. To assess the physical condition of participants before the start of the intervention, an adapted version of the International Physical Activity Questionnaires (IPAQ) was used. The SF-36 questionnaire was also used to measure the quality of life and energy/fatigue for participants. Twenty-four hours before moderate intensity exercise (MIE) and 48 h after MIE, participants underwent a fitness test that included a 6 min walking test distance (6WTD) and a spirometry test to assess aerobic capacity and endurance. A licensed nurse collected fasting blood samples before and after intervention. Blood pressure (fully automatic wrist blood pressure monitor), heart rate, and electrocardiograms (ECG) were measured by a research assistant before and after training and fasting intervention in a physiology lab at Qatar University.

The participants were randomized into two groups: the first group (*n* = 16) was the control group (4W), which underwent a 4-week exercise training program, and the second group (*n* = 13) underwent a 4-week exercise training program during Ramadan (4W + F) while fasting for 14 h (abstinence from food, drink, and smoking from sunrise to sunset).

### 2.2. Physical Activity Description

All qualifying participants underwent a moderate intensity exercise (MIE) program 3 times/week. Exercise timing for both groups was in the afternoon period (1–3 p.m.). The exercise training protocol included moderate (13–15 Borg scale, 50–70% VO_2_max) exercise training following the American Society of American College of Sports Medicine (ACSM)’s recommendations. The three sessions lasted 30 min each, with selected aerobic exercises based on walking and running. The level of exercises was based on the participants’ cardiorespiratory (heart and lung) and muscular fitness. 

The six-minute walking test distance (6WTD) is a submaximal test used to measure the distance that a participant can cover while walking on a flat, hard surface within a 6 min period.

Energy/fatigue was the primary endpoint and was assessed using the fatigue severity scale (FSS), which is a self-administered SF-36 questionnaire with nine questions. Every question is accompanied by statements that correspond to a score between 1 (strongly disagree) and 7 (strongly agree) [16].

The Borg scale test consists of the Borg rating of perceived exertion (RPE), which is an evaluation of exercise intensity and was determined for each participant after the 6WTD and after the training session. The Borg scale is a tool to measure a person’s perception of their effort and exertion, breathlessness, and fatigue during a workout.

To justify our choice of the 4-week training period, we took into consideration different variables: the progressivity of the load (i.e., frequency, volume, and intensity of exercise), skills acquisition, the monitoring and evaluation factor, behavioral changes, and the physiological adaptations to training in young recreational female athletes. Regarding the load, it was demonstrated that a minimum of 4 weeks (intensity (up to 66%) and volume (33–66%)) is required to maintain endurance adaptations [17]. The scientific literature suggests that a progressive, well-structured approach is required to improve muscular endurance and physiological hypertrophy in humans in the first 4 weeks of exercise [18]. Furthermore, previous studies have explored the rate of adherence during short- and long-term training periods and found that shorter periods (≤4 weeks) have higher rates of adherence compared to longer periods, especially if the sessions are well monitored [19,20]. In addition, the fasting protocol was also set to 4 weeks (the period during the month of Ramadan), not less and not more to keep the adherence rate high, as the subjects may be more prone to drop out if the study were a longer duration due to this intensive lifestyle intervention that combines fasting plus exercise training.

### 2.3. Clinical Trait Measurements

Blood samples were sent to a licensed medical laboratory to measure fasting blood sugar, HbA_1C_, total cholesterol, triglycerides, HDL, and LDL. Insulin levels were measured in serum samples using the Mercodia Insulin ELISA kit (Uppsala, Sweden), according to the manufacturer’s instructions. The absorbance was read using a Cytation5 imaging reader (BioTek, Washington, DC, USA). Body fat, fat free mass, fat mass, and muscle mass were measured using a TANITA body composition monitor. HOMA-IR was calculated using the following formula: HOMA-IR = fasting insulin (microU/L) × fasting glucose (mmol/L)/22.5.

### 2.4. Cytokine Profiling

The ProcartaPlex™ Human Mix & Match cytokine multiplex kit (MAN0024966, Invitrogen) was used to simultaneously profile cytokines, including IL-1 beta, IL-1RA, IL-6, IL-10, IL-22, MCP-1/CCL2, and TNF-α, using LUMINEX 200, according to the manufacturer’s instructions. Separate standard curves were used to validate the assay for the detection and quantification of cytokines using Xponent software (Version 6.2), according to the manufacturer’s instructions.

### 2.5. Antioxidant Enzyme Activity Measurements

The activities of superoxide dismutase and catalase were determined using colorimetric activity assays (EIACATC and EIASODC, respectively), according to the manufacturer’s instructions (ThermoFisher Scientific, Fredrick, MD, USA). The absorbance was read using a Cytation5 imaging reader (BioTek, USA). 

### 2.6. Measurement of Telomere Length

PureLink^®^ Genomic DNA Kits (Invitrogen, Life Technologies, Carlsbad, CA, USA) were used for the isolation of genomic DNA from the clotted blood at the bottom of the serum tubes, according to the manufacturer’s instructions. The concentration and quality of the DNA was measured using the Nanodrop. An Absolute Human Telomere Length Quantification qPCR Assay Kit (ScienCell, Carlsbad, CA, USA) used to measure the telomere length in extracted DNA samples, according to the manufacturer’s instructions. The kit includes a telomere primer set that amplifies telomere sequences, a single copy reference region for data normalization, and a reference genomic DNA sample with known telomere length as a reference for calculating the telomere length of target samples. Briefly, two qPCR reactions were prepared for each genomic DNA sample: one with telomere (TL) and one with single copy reference (SCR) primer stock solutions. qPCR reactions were prepared by adding a genomic DNA template (5 ng/µL) to the primer stock solution (TL or SCR) and GoldNStart TaqGreen qPCR master mix. qPCR was run using an initial denaturation of 95 °C for 10 min, followed by 32 cycles of denaturation at 95 °C for 20 s, annealing at 52 °C for 20 s, and extension at 72 °C for 45 s using StepOne™ Real-Time PCR System (ThermoFisher). For quantification of TL, ∆Cq (TL) was quantified by assessing the TL cycle number difference between the two genomic DNA samples (sample of interest and the reference genomic DNA sample with known telomere length). For SCR, ∆Cq (SCR) was assessed by quantifying the SCR cycle number difference between the two genomic DNA samples (sample of interest and the reference genomic DNA sample with known telomere length). ∆∆Cq was calculated as ∆Cq (TL) − ∆Cq (SCR). Fold change was assessed as 2^−∆∆Cq^, and the TL was expressed as a T/S ratio.

### 2.7. Statistical Analysis

Measurements were checked for normality using the Shapiro–Wilk test. Outliers were identified using the GOUT method, and after the inspection of the participants, the outliers were removed. A parametric paired *t*-test or a nonparametric Wilcoxon matched pairs signed rank test was performed accordingly. Spearman’s correlation was performed between the clinical measurements and the change in telomere length in the 4W + F group. All statistical tests were performed using the R statistical computing language (version 4.2.1). Figures were generated using GraphPad Prism (v. 10.1.0) and Cytoscape (v. 3.9.1).

## 3. Results

### 3.1. Baseline Characteristics of Participants

Table 1 provides an overview of the baseline characteristics of the participants in this study, which are grouped into 4W and 4W + F groups. Notably, most of the parameters exhibited no significant differences between the two groups, except for the 6WT distance and HDL.

### 3.2. Telomere Length before and after Intervention in 4W and 4W + F

The paired comparison of telomere length before and after intervention in 4W participants (Figure 1A) and in 4W + F participants (Figure 1B) showed that 4 weeks of MIE exhibited no significant change in TL before and after training (Table 2). Whereas, when participants underwent 4 weeks of training along with fasting, there was a significant (*p* = 0.048) increase in TL (Table 2). Appendix A reveals all changes triggered by intervention in both groups.

Changes in the T/S ratio in the 4W and 4W + F groups were assessed using the Wilcoxon matched pairs test/paired *t*-test according to a normality test.

### 3.3. Comparing the Percentage Change in Participants’ Markers in 4W versus 4W + F

The percentage change before and after exercise was compared between the two study groups using regression analysis. The results (Table 3) showed that the percentage change in telomere length (*p* = 0.021) was significantly higher and that the percentage change in TNF-α (*p* = 0.013) was significantly lower in the 4W + F group compared to the 4W group. The estimate reflects the percentage change in 4W + F compared to 4W.

### 3.4. Correlation of Telomere Length Change with Percentage Change in Participants’ Markers

In order to understand the potential implications of TL change on the general health of active fasting individuals, Spearman correlation was performed for TL change before and after intervention and the changes in other clinical parameters and biomarkers. The results showed that the TL change correlated positively with the change in HDL in the 4W + F group (Figure 2). Correlations between TL change and other clinical parameters were not significant.

## 4. Discussion

Developing effective interventions to maintain health and well-being and to prevent or reduce the burden of age-related diseases is a goal of public health. Telomeres play a crucial role in cellular aging and health. Telomere length can be influenced by various lifestyle factors. Dietary restrictions and activities have the potential to reduce the rate of telomere shortening, to prevent telomere attrition, or even to increase the length of telomeres, leading to a delayed onset of age-associated diseases and increased lifespan [21]. In this longitudinal study, we investigated the combined effect of exercise and fasting on telomere length in healthy, young, non-obese females.

Our emerging results showed that TL increased significantly from baseline after the intervention in the 4W + F group, but not in the 4W group. Moreover, a significant difference was recorded when we compared the mean percentage change in TL of participants in the 4W + F group versus the 4W group.

Several studies have produced evidence that telomere lengthening occurs in several species [22,23,24]. Ornish et al. [25] were the first to report a significant increase in relative TL after intervention in humans. Thereafter, two studies [26,27] reported telomere elongation in 16.7% and 6.3% of their study participants after follow-up. Ornish et al. showed that TL increased by a median of 0.06 T/S units in the lifestyle intervention group when compared to controls, which showed a decrease in TL. However, the increase in TL in their study was observed after 5 years of follow-up. Interestingly, positive changes in relative TL in our study were detectable in a short period.

Collective evidence suggests that exercise and fasting may independently contribute to changes in telomere length. Although they did not report an increase in TL over time, many of these studies showed a higher TL in athletes compared to inactive control subjects [28,29,30], and that fasting may prevent telomere shortening [31], especially in women [32,33]. On the other hand, our previous study showed a significant increase in TL with sport intensity, especially in the younger group [34]. Additionally, Brandao et al. recorded an increase in TL of 2% after 8 weeks of exercise training in obese women [35], and Garcia-Calzon et al. reported that a two-month energy-restricted diet resulted in an increased TL among obese adolescents [36]. However, different results were published by Shin et al. [37] and Nickels et al. [38], who reported that exercise training was associated with no changes in TL. Moreover, a systematic review suggests that there is no effect of diet on TL [39], and a randomized controlled trial concluded that dietary weight loss with exercise did not change TL in postmenopausal women after a 12-month follow-up [40]. It is important to note that longitudinal studies have suggested that an increase in relative TL is mainly due to measurement error [41]. We, however, recruited the 4W control group and the 4W + F group using the same selection criteria and conducted blinded laboratory assessments. Therefore, any sources of measurement error would probably have affected both groups equally. It is also noteworthy to mention that individual telomere biology is highly dynamic and could be largely influenced by the individual’s genetic background.

Our emerging data also showed a significant positive correlation between TL and HDL in the 4W + F group only. A positive association between TL and HDL was previously reported in many studies [27,42,43]. This association might be explained by oxidative stress and inflammation status. Indeed, HDL possesses anti-inflammatory properties and protects against oxidative stress; on the other hand, inflammation and oxidative stress have been linked to TL [43]. Studies have demonstrated that oxidative stress is a major factor that significantly contributes to the rate of telomere shortening [44], and chronic inflammation, characterized by elevated levels of reactive oxygen species, revealed shortened telomeres in inflamed tissues compared to normal tissues [45]. Oxidative stress can lead to the accumulation of modified bases within telomeres, which inhibits telomerase activity and ultimately disrupts TL [46].

Our findings were further corroborated by the significantly lower levels of TNF-α in the 4W + F group compared to the 4W group. TNF-α is a pro-inflammatory cytokine that plays a significant role in the body’s inflammatory response and a crucial role in inducing oxidative stress [47]. Various studies have shown that both psychological and physical stress can lead to an increase in the production of TNF-α [48]. Decreased TNF-α levels were shown to be associated with Ramadan fasting [49] and exercise [50,51]. Interestingly, TNF-α has an inverse relationship with TL, and TNF-α-induced telomere shortening was previously reported [52,53,54]; this shortening was observed to be mediated by the stress-responsive transcription factor ATF7 [53]. Moreover, TNF-α was reported to be associated with decreased telomerase activity [55].

Exercising while fasting has been shown to provide many benefits, including improved lipolysis and fat oxidation, prioritized muscle repair over autophagy, and improved mitochondrial respiratory efficiency [56]. While exercise has been shown to increase telomerase activity [15,57], the positive effect of dietary restriction on telomerase activity or its expression is less clear. Unfortunately, measurements of telomerase activity were not included in our study, and we acknowledge it as a limitation of the study. Another limitation is the relationship between TL and aging. While TL is a prominent biomarker for aging, we agree that aging is an extremely complex phenomenon that cannot be accurately measured with a single biomarker. Moreover, our study measured only leukocyte TL, which may not be representative of most tissues and organs or the progression of most age-related diseases; however, we would like to emphasize that peripheral leukocytes are a readily accessible and dynamic cell population that can provide valuable insight into systemic aging processes.

It is important to note that the fasting period remains consistent for all participants during Ramadan, which is a deliberate choice to ensure the uniformity of fasting times among all participants and to guarantee compliance. Although all participants were encouraged to maintain a balanced diet, they did not adhere to a specific, predefined dietary regimen, and this could be considered as an additional limitation of our study. Moreover, the absence of a Ramadan fasting group without exercise and the absence of a control group without exercise add to the limitations. A replication of the study with a larger and different paired age population (from 20 to 80 years old) is warranted to increase the generalizability of the findings and for definitive recommendations to be made. Moreover, a crossover design in which the two groups undergo the same procedure can facilitate the examination of transient or permanent changes in telomere lengths, as well as cellular aging. Future recommendations also include expanding the study to include varying degrees of fasting.

## 5. Conclusions

To the best of our knowledge, this study is among the first to report an increase in telomere length after combining Ramadan fasting with training, suggesting that exercising while fasting may be an effective tool for slowing down the aging rate. We hypothesize here that combining fasting with exercise may have additive or synergistic effects and could cause unforeseen changes that are different from those seen during separate bouts of fasting or exercise, and that TNF-α and HDL contribute directly or indirectly to the observed changes in telomere length through their opposing roles in inflammation and oxidative stress. However, the exact mechanisms by which reduced inflammation and oxidative stress lead to increased telomere length are not yet fully understood, and more in-depth studies involving measurements of telomerase activity are needed to determine the underlying effects. 

## Figures and Tables

**Figure 1 biomedicines-12-01182-f001:**
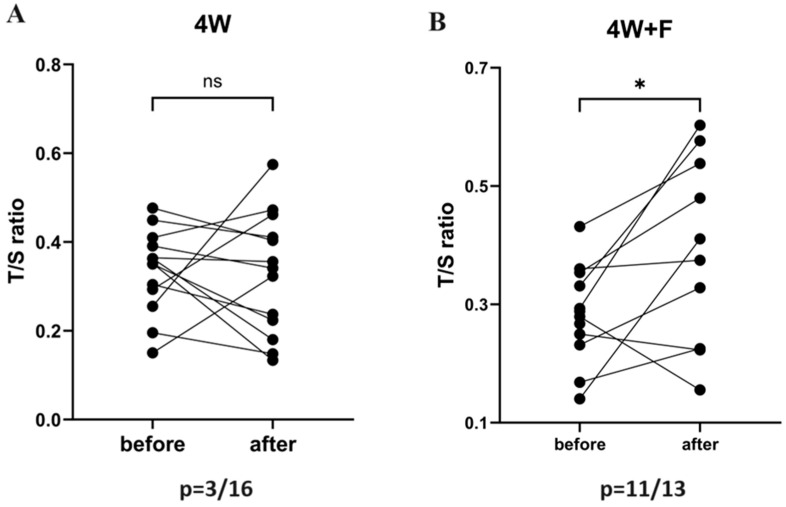
Comparing paired TL before and after exercise in (**A**) 4W and (**B**) 4W + F. * and ns indicate significant and non-significant changes respectively. *p* represents the proportion of participants who exhibited an increase in the T/S ratio after the intervention.

**Figure 2 biomedicines-12-01182-f002:**
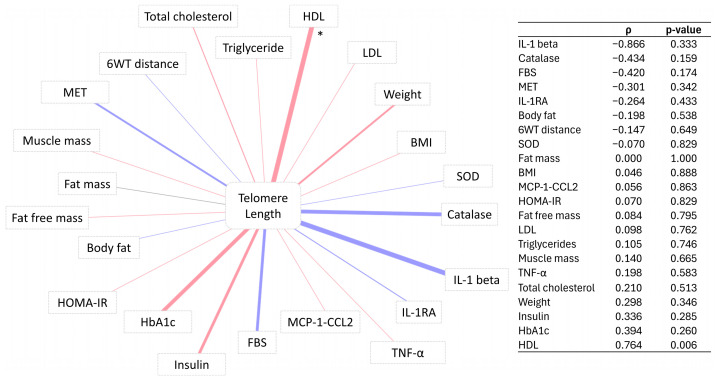
Network of correlation between clinical traits and telomere length in the 4W + F group. Red and blue reflect the positive and negative Spearman’s ρ, respectively. * signifies *p*–value < 0.05. The adjacent table displays the associated Spearman’s ρ and *p*–values.

**Table 1 biomedicines-12-01182-t001:** Baseline characteristics of participants.

	4W (*n* = 16)	4W + F (*n* = 13)	*p*-Value
Age	21 (20–26)	22 (20.75–22.25)	0.700
BMI	24.78 (5.87)	24.42 (4.54)	0.862
TL (T/S ratio)	0.33 (0.09)	0.28 (0.08)	0.153
Height (m)	1.6 (0.06)	1.59 (0.05)	0.611
Weight (kg)	65.1 (49.1–71.3)	58.75 (51.8–65.72)	0.913
Body fat (%)	0.31 (0.11)	0.32 (0.09)	0.659
Fat free mass (kg)	42.95 (4.65)	43.85 (4.49)	0.629
Fat mass (kg)	20.54 (11.68)	21.86 (11.36)	0.777
Muscle mass (kg)	40.77 (4.43)	41.59 (4.27)	0.641
MET	975 (622–1480)	1251.25 (826.5–2708.25)	0.265
6WT Distance (m)	531 (62.71)	587.83 (70.68)	0.045
Handgrip (L)	21.83 (4.04)	21.92 (5.6)	0.966
Handgrip (R)	23.44 (5.1)	22.18 (5.37)	0.556
Insulin (mU/L)	13.03 (7.69)	12.13 (5.13)	0.731
FBS (mmol/L)	5.1 (4.7–5.3)	5.3 (4.8–5.32)	0.284
HOMA-IR	2.39 (1.76–3.94)	3.73 (1.99–4)	0.496
Total cholesterol (g/dL)	184 (159–202)	161.5 (147.75–174.5)	0.103
Triglycerides (g/dL)	64 (50–75)	59.5 (48.5–66)	0.549
HDL (g/dL)	68.31 (13.81)	55.45 (14.7)	0.040
LDL (g/dL)	101.46 (24.37)	101.92 (35.19)	0.971
HbA_1C_ %	4.9 (4.7–5.5)	5.25 (4.82–5.38)	0.935
SOD (U/mL)	0.97 (0.87–1.22)	0.92 (0.83–1.08)	0.463
Catalase (U/mL)	19.61 (18.78–19.92)	20.09 (18.69–20.21)	0.586
IL-1 beta (pg/mL)	0.14 (0.14–0.14)	0.14 (0.14–0.32)	0.210
IL-6 (pg/mL)	24.77 (24.77–24.77)	24.77 (24.77–30.68)	0.765
IL-8 CXCL8 (pg/mL)	1.37 (0.83)	2.4 (1.72)	0.195
IL-1RA (pg/mL)	294.02 (176.92–607.44)	457.1 (204.11–1068.14)	0.518
TNF-α (pg/mL)	0.43 (0.43–3.95)	0.43 (0.43–3.95)	0.635
MCP-1 (CCL2) (pg/mL)	345.82 (226.46)	271.42 (172.68)	0.363
Physical function	0.69 (0.26)	0.77 (0.25)	0.471
Physical health limitation	0.75 (0.5–1)	0.62 (0.25–1)	0.932
Emotional problems limitation	0.67 (0.33–1)	0.5 (0–1)	0.632
Energy/Fatigue	0.55 (0.2)	0.54 (0.1)	0.851
Emotional well being	0.72 (0.64–0.76)	0.66 (0.58–0.72)	0.459
Social function	0.58 (0.24)	0.6 (0.23)	0.846
Pain	0.76 (0.23)	0.64 (0.28)	0.255
General health	0.71 (0.18)	0.66 (0.12)	0.422
Health change	0.75 (0.75–1)	0.75 (0.5–0.75)	0.141

Data are presented as mean (SD)/median (IQR) for parametric and nonparametric variables. Student’s *t*-test and Mann–Whitney U test were performed accordingly.

**Table 2 biomedicines-12-01182-t002:** Paired test of TL (expressed as T/S ratio) in 4W and 4W + F groups.

Group	Before	After	*p*-Value
4W	0.335 (0.094)	0.328 (0.137)	0.87
4W + F	0.284 (0.24–0.34)	0.393 (0.22–0.55)	0.048

**Table 3 biomedicines-12-01182-t003:** Percentage change comparison in participants’ markers between 4W and 4W + F.

Percentage Change	Estimate	SE	*p*-Value
TNF-α	−568.854	215.301	0.013
Telomere length	113.493	46.995	0.021

## Data Availability

The datasets used and/or analyzed during the current study are available from the corresponding author on reasonable request.

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
