# Peer review of "Joint Effects of Exercise and Ramadan Fasting on Telomere Length: Implications for Cellular Aging"

_biomedicines, 2024, doi:10.3390/biomedicines12061182_

Round 1
Reviewer 1 Report
Comments and Suggestions for Authors
General comments
The authors have quantified leukocyte telomere length in a small cohort of young females that either exercised or exercised during the intermittent fast associated with Ramadan. Data from individuals that were not exercised were not included, nor were data from individuals that had only fasted. Nevertheless, there is the suggestion that exercise alone had no significant effect on telomere length while exercise in addition to fasting did have a significant effect. There was also an effect on circulating TNF-alpha level, but no other pro-inflammatory cytokines seem to have been affected. This is surprising. The notion of using the intermittent fasting associated with Ramadan is a novel approach and should result in strong adherence to the fasting protocol, which is always questionable in human studies. Also, because of the variance in fasting duration that occurs “naturally” due to latitudinal variation among populations practicing Ramadan, there is the possibility of expanding this study design to include those undergoing varying degrees of fasting.
Specific comments
1. More information about the duration of the fast, as well as the “rules” of the fast for those that may be unfamiliar with Ramadan fasting would be helpful, especially the restriction against ingesting fluids.
2. When did the training (exercise) sessions take place (e.g, morning, evening) for both groups? For the fasted group, would this be at the end or the beginning of the fast?
3. Why weren’t the data transformed to meet the normality assumption rather than just removing outliers? Also, why were both parametric and non-parametric tests used if the outliers were removed?
4. For the data in Table 1, why weren’t these measures repeated after the end of the training/exercise? It is labeled as baseline which suggests these data were collected prior to training.
5. The 4W+F group already walked significantly greater distance than the 4W group prior to training? Isn’t this a confounding factor?
6. In lieu of, or maybe in addition to, the individual T/S ratios in Figure 1, it would be useful to report the proportion of individuals within each group that exhibited an increased in telomere length. Perhaps on the figure itself.
7. For the data in Figure 2, where are the percentage changes for the individual markers reported? This would be useful. Also, these data are better presented in a tabulated form rather than a figure. A series of lines of varying thickness is not intuitive.
8. Caloric restriction and fasting are not physiologically equivalent.
Reviewer 2 Report
Comments and Suggestions for Authors
The MS described the 4-week exercise and ramadan fasting on telomere length. For making a conclusion for effects of exercise and ramadan fasting on telomere length, it was suggested to include different paired age populations (from 20-year-old to 80- year-old) and increased participants of each group.
1. Please explain the reason for the 4-week exercise rather than other time intervals (such as 2-week? 3-week? or 5-week?).
2. It was suggested to crossover two group for the same procedure in order to check the transit or permanent changes of telomere lengths, and also for cellular ageing.
Round 2
Reviewer 1 Report
Comments and Suggestions for Authors
The authors have adequately addressed my concerns. I have no further comments.
Reviewer 2 Report
Comments and Suggestions for Authors
It is acceptable in the present format.